# Optimal Volume Assessment for Serous Fluid Cytology

**DOI:** 10.3390/biomedicines12040899

**Published:** 2024-04-18

**Authors:** Konstantinos Christofidis, Maria Theochari, Stylianos Mavropoulos Papoudas, Lamprini Kiohou, Stylianos Sousouris, Areti Dimitriadou, Nikolaos Volakakis, Nicoletta Maounis, Panagiota Mikou

**Affiliations:** 1Cytopathology Laboratory, “Laiko” General Hospital of Athens, 11527 Athens, Greece; 2Oncology Unit, “Hippokration” General Hospital of Athens, 11527 Athens, Greece; 3Department of Biology, University of Crete, 70013 Heraklion, Greece; 4Cytopathology Laboratory, “Sismanoglio-Amalia Fleming” General Hospital of Athens, 15127 Athens, Greece

**Keywords:** serous effusion cytology, optimal volume, The International System for Reporting Serous Fluid Cytopathology (TIS)

## Abstract

Objective: This study aimed to investigate the optimal volume of serous fluid needed for accurate diagnosis using The International System for Reporting Serous Fluid Cytopathology (TIS), as well as to provide information on the distribution of serous effusion cases in the TIS categories (ND: non-diagnostic, NFM: negative for malignancy, AUS: atypia of undetermined significance, SFM: suspicious for malignancy, MAL: malignant) and relevant epidemiological data. Methods: A retrospective analysis of 2340 serous effusion cases (pleural, peritoneal, and pericardial) from two hospitals between 2018 and 2020 was conducted. TIS categories were assigned to each case, and for 1181 cases, these were correlated with the volume of the analyzed fluid. Results: Our study found statistically significant differences in volume distributions between certain TIS categories. Statistically lower volumes were observed in NFM compared to MAL, in UNCERTAIN (ND, AUS, SFM) compared to both MAL and NFM, and in NOT MAL (ND, NFM, AUS, SFM) compared to MAL. However, these differences were not substantial enough to hold any clinical relevance. Conclusions: This study suggests that while fluid volume may slightly influence the TIS category, it does not impact the diagnostic accuracy of serous effusion cytology. Therefore, the ideal serous effusion specimen volume can be defined solely by practical parameters.

## 1. Introduction

Serous effusions can arise from a variety of neoplastic and non-neoplastic disorders [1,2,3]. Cytological examination of an effusion specimen is a key step in differentiating between these two categories [4,5,6]. By combining the morphological characteristics of a malignant sample with immunocytochemical (ICC) findings, an experienced cytopathologist can not only diagnose malignancy but also precisely locate the organ of origin, providing valuable information regarding the extent of the disease and aiding in the patient’s clinical management [4,5,6]. However, there is a gray zone of cytology samples with atypical characteristics that lie between clearly benign and overtly malignant cases, the significance of which is often difficult to determine.

Following the successful implementation of the Bethesda System for cervical cytology [7], the Paris System for Reporting Urinary Cytopathology [8], the Bethesda Classification of Thyroid Nodule Fine Needle Aspirations [9], and other organ-specific cytopathology classifications [10,11,12], it has become apparent that a similar system should be applied for reporting pleural, peritoneal, and pericardial effusion specimens. The endorsement and application of a universal reporting system for serous effusion cases would offer numerous potential benefits, including better communication and understanding amongst cytopathology laboratories, as well as between laboratory scientists and clinicians, aiding in the amelioration of patient clinical management based on the cytology report [13].

Considering the above, a collaboration of scientists from the International Academy of Cytology (IAC) and the American Society of Cytopathology (ASC) developed a system of classification with clearly defined criteria for each diagnostic category [14]. Thus, the International System for Reporting Serous Fluid Cytopathology (TIS) has been introduced, which classifies serous effusion samples into five diagnostic categories: non-diagnostic (ND), negative for malignancy (NFM), atypia of unknown significance (AUS), suspicious for malignancy (SFM), and malignant (MAL) [14]. The utility of the proposed classification has been widely reported [15,16,17,18], and its global impact has been reviewed [19]. 

In this study, we aimed to evaluate the optimal volume of fluid to be assessed, as this has not been extensively studied. We retrospectively gathered all pleural, peritoneal, and pericardial effusion cases from two hospitals in Greece for a 3-year period. We categorized each specimen according to TIS and then correlated each report with the patient’s following or concomitant histological report or clinical follow-up. The volume of the fluid analyzed was recorded, and its effect on the diagnostic yield was assessed.

## 2. Materials and Methods

### 2.1. Patients’ Selection

The study protocol received approval from the Institutional Ethical Board Committees of both “Laiko” and “Sismanoglio-Amalia Fleming” General Hospitals of Athens. The cytopathology departments’ databases of these two tertiary hospitals were queried to retrieve all serous effusion cases between 2018 and 2020. Our search retrieved 2340 specimens. Data collected for each case included patient demographics (gender and age), sample volume (available for 1181 specimens), and the number of immunocytochemical studies performed on each sample. Subsequently, the histopathology and oncology department databases were accessed to obtain comprehensive information on the follow-up of each patient.

### 2.2. Specimen Processing

Standard serous effusion specimen processing in both the laboratories involves centrifugation (Thermo Scientific SL 16, Thermo Electron LED GmbH, Osterode, Germany) and preparation of two conventional smears: one fixed in ethanol for Papanicolaou staining and another air-dried for Giemsa staining. The cytotechnologists then proceed to the liquid-based preparation of the sample (ThinPrep2000 Processor, Cytyc Co., Boxborough, MA, USA), where cellular morphology can be more clearly examined, and potential ICC and molecular studies can be conducted. Cell block preparation is occasionally employed for complex cases requiring extensive ICC evaluation. Notably, both departments are staffed by specialized cytopathologists, and consensus between two cytopathologists is mandatory for issuing reports in particularly challenging cases.

### 2.3. Reports’ Categorization

Reports were categorized according to The International System for Reporting Serous Fluid Cytopathology, and slides with indeterminate diagnoses were re-evaluated by PM and NM. The initial cytology reports proved invaluable, as, in most cases, the extensive and analytical description of the cytomorphological characteristics of the sample were sufficient to roughly classify each specimen into one of the five following TIS categories:ND: Non-diagnostic specimen.NFM: Specimens with clearly benign characteristics.AUS: Specimens containing cells exhibiting some degree of atypia, lacking definitive features of malignancy, and often leaning towards benignity with atypia attributed to inflammation.SFM: Cases with atypical cells strongly resembling malignant ones, but without enough atypia or a sufficient number of atypical cells to warrant a malignant diagnosis.MAL: Specimens containing unequivocally malignant cells.

### 2.4. Statistical Analysis

For all statistical analyses, the software used was the R version 4.1.2 (2021-11-01)—“Bird Hippie”, Copyright (C) 2021, The R Foundation for Statistical Computing, Platform: x86_64-pc-linux-gnu (64-bit). 

## 3. Results

### 3.1. Pleural Effusions

A total of 1594 pleural effusion cases were identified in both departments (Table 1). The male-to-female ratio and average age were 1.48 and 70.92 years, respectively. Immunocytochemistry was performed in 176 cases (Table 2). Reclassification based on the TIS guidelines resulted in the redistribution of 66 cases (4.14%) to the ND category, 1228 (77.04%) as NFM, 39 (2.45%) as AUS, 51 (3.2%) as SFM, and 210 (13.17%) as MAL (Table 1). The epidemiological data on the pleural effusion specimens is summarized in Table 2. Lung carcinoma was the most common metastatic tumor type, followed by breast carcinoma and mesothelioma (Figure 1).

### 3.2. Peritoneal Effusions

A total of 700 reports pertained to peritoneal effusion specimens. The male-to-female ratio and the average age were 0.98 and 67.6 years, respectively, and ICC was performed in 176 cases (Table 3). Of these cases, 15 (2.14%) were classified as ND, 484 (69.14%) as NFM, 21 (3.0%) as AUS, 20 (2.86%) as SFM, and 160 (22.86%) as MAL (Table 1). The epidemiological data on the peritoneal effusion specimens is summarized in Table 3. Ovarian carcinoma was the most common cause of peritoneal malignant effusions, followed by stomach, breast and colon neoplasms (Figure 2).

### 3.3. Pericardial Effusions

A total of 46 cases of pericardial effusions were identified, with a gender ratio and average age of 1.42 and 60.78 years, respectively (Table 4). We classified 5 cases (10.87%) as ND, 27 (58.7%) as NFM, 1 (2.17%) as SFM, and 13 (28.26%) as MAL (Table 1). The epidemiological data on the pericardial effusion specimens is summarized in Table 4. The lung was the organ of origin for the four cases with existing histopathology reports proving the presence of malignancy.

### 3.4. Optimal Volume Assessment

Pleural, peritoneal, and pericardial samples were pooled for analysis of optimal volume for diagnosis. All serous effusion samples with volume data were initially divided into three oncometric subgroups: small (<10 mL), medium (10–500 mL), and large (>500 mL). This arbitrary categorization aimed to provide clinically significant conclusions. The numbers and percentages of each TIS category for each subgroup were assessed. Volume data were available for 1181 out of the 2340 specimens. There were 367 specimens with volumes less than 10 mL, 725 specimens with volumes between 10 and 500 mL, and 19 specimens with a volume greater than 500 mL. The distribution of the five TIS diagnostic categories across the three volume groups is shown in Table 5.

When closely looking at the table, the values presented appear to show higher malignancy rates for specific volumetric bins for specific groups. Notably, volumes greater than 10 mL seem to have a higher malignancy rate than those less than 10 mL. Moreover, the percentages of ND and SFM are higher in the small-volume group compared to the medium-volume group. Additionally, the percentage of AUS in the large-volume group was more than double that in the medium-volume group. Medium volumes (10–500 mL) of specimen fluid resulted in fewer nondiagnostic and indeterminate (AUS and SFM) diagnoses. To assess whether this has statistical significance a statistical test was applied. Since the chi-squared test approximation may be unreliable for contingency tables containing values less than five, we used the Fisher’s exact test to get a more accurate result. For this contingency table, a Fisher’s exact test for count data will give the *p*-value of 0.07828, which is not statistically significant.

Following this, the distribution of the volumes was studied, and the outliers (absolute modified z-score ≥ 3) were removed. Nevertheless, the volumes were still not normally distributed (Shapiro test, *p*-value < 2.2 × 10^−16^). The comparative boxplots of the volumes are shown in Figure 3 and the distribution of cases according to TIS in Table 6 (both with outliers removed). The circles outside the boxplot’s whiskers in all the following figures represent values outside 1.5 × interquartile range.

A Kruskal–Wallis non-parametric ANOVA was performed to check for differences between the distributions, which were statistically significant (*p*-value = 0.01001292). Dunn’s test with Bonferroni adjustment was performed to isolate the groups which present differences (Table 7). Interestingly, the significant difference lies between MAL and NFM, whose distributions are summarized in Table 8. NFM is shown to have statistically significantly lower values/volumes than MAL. A plot showing the distributions of the MAL and NFM Volumes can be seen in Figure 4.

In the next step of the analysis, TIS were divided into three categories: MAL, NFM, and UNCERTAIN (namely the indeterminate diagnoses: ND, AUS, and SFM), and the analysis was repeated. The Kruskal–Wallis test was significant (*p*-value = 0.003053868). Dunn’s test showed that it is possible to differentiate between MAL and NFM, as well as between MAL and UNCERTAIN based on the volume distributions; however, it is impossible to differentiate between NFM and UNCERTAIN (Table 9). The comparative summary distributions of the former (Table 10) as well as the comparative boxplot (Figure 5) demonstrate that UNCERTAIN tends to have lower volume than both MAL and NFM, although the difference with NFM is marginal and not significant.

Finally, the dataset was segmented into two parts, MAL (151, 15.5%) and NOT MAL (including ND, NFM, AUS, SFM. 826, 84.5%), in an effort to isolate the differences between volumes. This difference was also statistically significant (Mann–Whitney U test, *p*-value = 0.001408). The NOT MAL samples appear to have slightly lower volume than MAL samples (Figure 6, Table 11).

Concerning the size of the sample, in this study, we use the Kruskal–Wallis test, which is a non-parametric ANOVA test. We elected to use Cohen’s f statistic as the effect size index to use for our Kruskal–Wallis analysis of variance. That way we can measure a standardized average effect in our observations across all the levels of the volume variable. The calculated value for our dataset was f = 0.1248913. According to Jacob Cohen, the values of 0.10, 0.25, and 0.40 represent small, medium, and large effect sizes, respectively [20]. The balanced one-way analysis of variance power calculation for significance level 0.05 outputs a power level of 1, denoting a very high confidence level for our sample size of 977 observations (outliers removed). Therefore, it appears that our sample size has adequate power to detect a small effect size, which is a strong indication that the study design is well suited to identify the effect under investigation.

## 4. Discussion

The optimal specimen volume for serous effusion cytology represents a pivotal concern, extensively explored by numerous researchers. This study aimed to contribute to this ongoing discussion and the development of consensus guidelines. Furthermore, we addressed the imperative need for internal hospital recommendations to guide clinicians.

Our investigation encompassed 2340 specimens over a three-year period from 2 hospitals, with oncometric data being available for 1181 specimens. Pleural effusions constituted the majority of cases, followed by peritoneal effusions and a smaller number of pericardial effusions. Notably, we observed a higher percentage of indeterminate (AUS and SFM) diagnoses in both low- and high-volume groups compared to the medium-volume group, as well as an overrepresentation of inadequate samples in the low-volume group. However, these observations were not supported by the subsequent statistical analysis. 

Aiming to correlate serous effusion samples’ volumes with the diagnostic categories of TIS, we began our analysis by dividing the samples into three volumetric groups, whose distributions showed no statistical significance. Consequently, we checked for differences between the distributions of the volumes of TIS, which showed a significant difference between MAL and NFM, where NFM proved to have statistically significantly lower volumes than MAL. Following that, we repeated the analysis, dividing our dataset into MAL, NFM and UNCERTAIN. We found out that UNCERTAIN have lower volume than MAL and NFM, although only marginally so in the latter case. Finally, we segmented the dataset into MAL and NOT MAL and ran a final analysis that demonstrated a statistically significant lower volume of the NOT MAL compared to the MAL samples. However, although some statistically significant differences in the volume distributions were observed, none of them were large enough to possess any clinically significant value. 

Our results are similar to those reported by Sallach et al., who conducted a retrospective study on the pleural effusions of 282 patients, using different volume thresholds [21]. Similarly, Abouzgheib et al. prospectively studied a group of pleural effusions from 44 patients and found no difference in the malignancy rate in groups under and over 50 mL [22]. Wu et al., in their study of 74 cases, failed to find any statistically significant difference in malignancy detection among 25 mL, 50 mL, and 150 mL sample volumes [23]. Additionally, Torous et al., with a cohort of 226 pleural fluid cases, did not conclude on an ideal volume [24]. 

On the contrary, Coconubo et al., in a study of 8530 serous effusions, claimed that fluid volume affects adequacy and detection of malignancy and reported 75–100 mL of fluid as the optimal volume for cytology [25]. Similarly, Beg et al., in a study of 1597 samples divided into 6 groups with different cutoff volume values, proposed 70 mL as the overall optimal specimen volume [26]. They also diagnosed malignancy in serous effusions of extreme volume sizes, from 5 up to 5000 mL, which is consistent with our findings. 

Rooper et al., in a retrospective study of 2540 pleural fluids, suggested an optimal volume of 75 mL [27]. Jha et al. have studied 939 pleural fluid cases and concluded on an ideal volume of 13.5 mL [28]. In another retrospective study on pleural effusions, Thomas et al. suggested a minimum volume of 25 mL [29]. Swiderek et al. conducted a prospective study on 120 pleural effusion patients and found the optimal volume to be 60 mL [30]. 

Zhang et al., in their study of 123 ascitic fluid patients, reported a cutoff value of 200 mL [31], while Rooper et al., with a cohort of 2665 patients with ascites, proposed a volume of 80 mL [32]. For pericardial effusions, Rooper et al. have considered 60 mL as optimal [33], while Dragoescu et al., with a cohort of 128 pericardial effusion patients, with a mean volume of 60 mL and a malignancy rate of 24.2%, did not find an optimal volume [34].

Our results, encompassing all types of effusions, indicate that malignancy can be detected in volumes less than 10 mL at a rate of 11.4%, while in medium- and large-volume groups, the rates are 17.7% and 17.9%, respectively. Malignancy detection does not proportionally increase with volume. Small volume differences do not significantly impact serous effusion cytology sensitivity. However, managing large volumes presents challenges in transport, handling, and storage for the hospitals. Medium-volume specimens show an equivalent rate of malignancy detection, lacking the aforementioned difficulties. In our study material, the mean volumes presented in Table 8, Table 10 and Table 11 range between 13.79 and 18.94, leading to the conclusion that the medium-volume group consisted mostly of specimens with a volume below 100 mL.

Our study has limitations due to its retrospective nature and the size of the dataset. Broad categorization of serous fluids might be considered a disadvantage compared to a more detailed analysis of multiple small-volume groups. Nevertheless, we chose this strategy in order to obtain an objective, clinically useful deduction. Another limitation of this study is that we have not statistically correlated patients’ age, gender, and prior history with the volumetric bins initially employed in our analysis.

## 5. Conclusions

In conclusion, rather than prescribing a strict optimal volume for serous effusion cytology, it may be more practical to recommend that the clinicians adhere to a guideline suggesting a medium-volume range (10–100 mL). Clinicians should be mindful that excessively large volumes do not enhance malignancy detection rates. On the other hand, in cases where only very limited samples can be acquired, cytological analysis should be nevertheless carried out. It seems that the ancient Greek quote “(πᾶν) μέτρον ἄριστον”, signifying moderation in all things, is aptly applicable to serous effusion cytology.

## Figures and Tables

**Figure 1 biomedicines-12-00899-f001:**
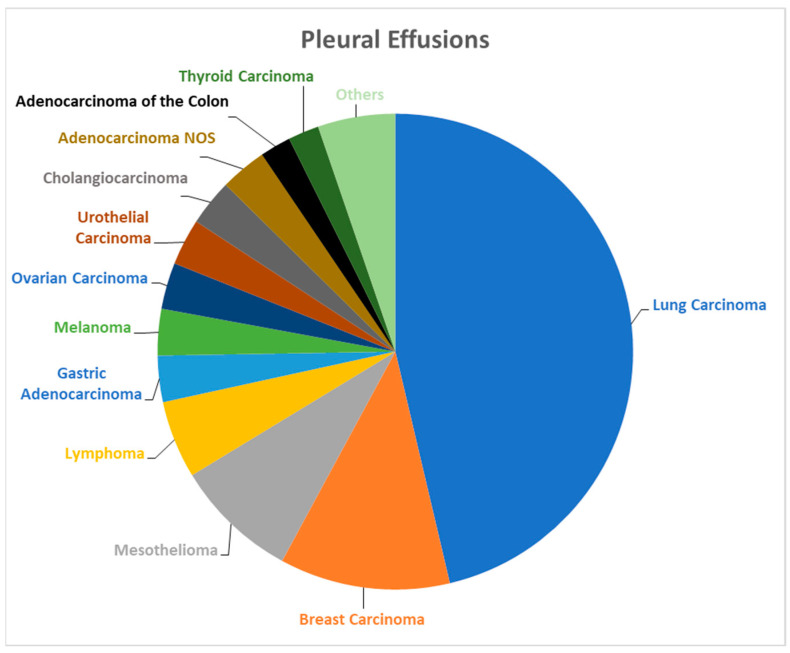
Malignant pleural effusions: tumor type/origin. The most common site of origin/type of neoplasm is lung carcinoma, followed by breast carcinoma, mesothelioma, and lymphoma.

**Figure 2 biomedicines-12-00899-f002:**
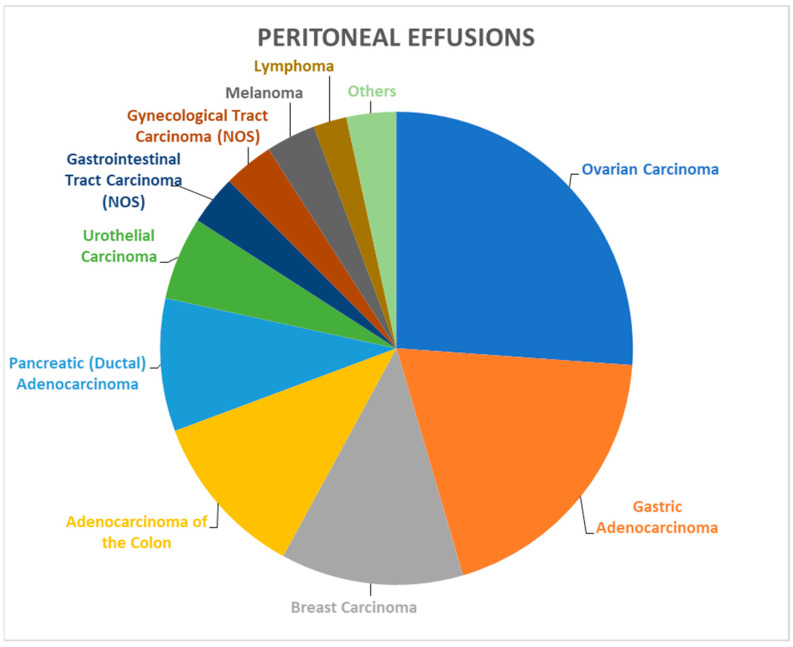
Malignant peritoneal effusions: tumor type/origin. The most common site of origin/type of neoplasm is ovarian carcinoma, followed by stomach, breast and colon adenocarcinomas.

**Figure 3 biomedicines-12-00899-f003:**
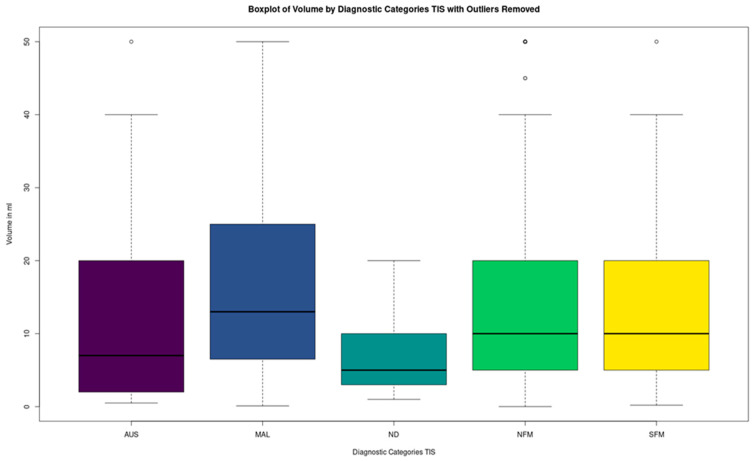
Boxplots of TIS~volume.

**Figure 4 biomedicines-12-00899-f004:**
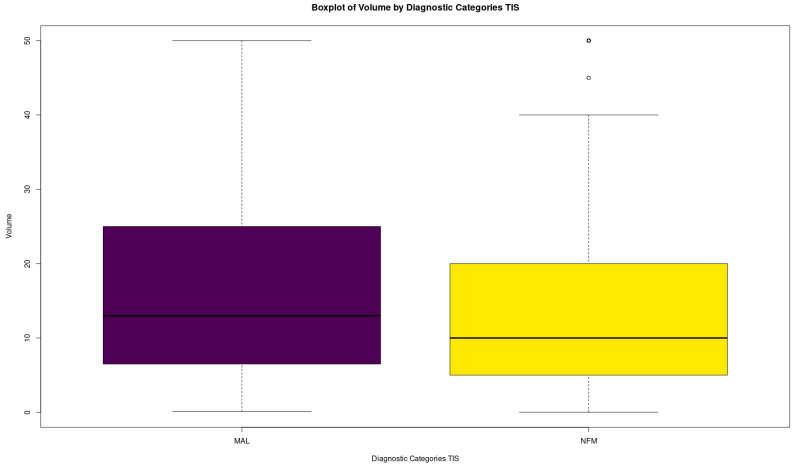
Boxplots of TIS~volume for MAL and NFM.

**Figure 5 biomedicines-12-00899-f005:**
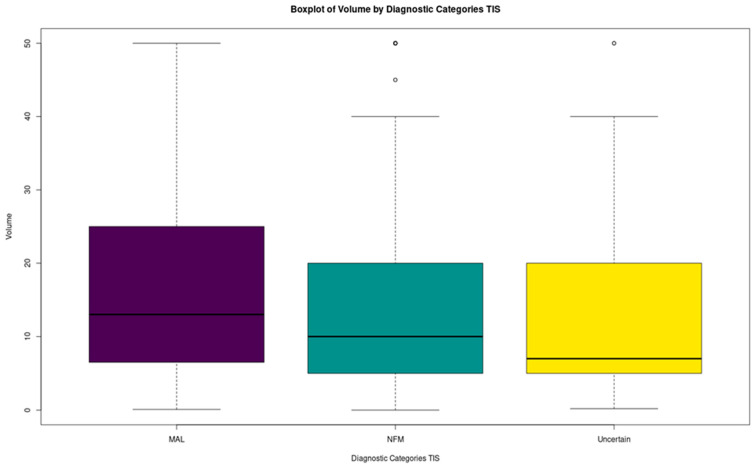
Boxplots of TIS~volume for MAL, NFM, and UNCERTAIN.

**Figure 6 biomedicines-12-00899-f006:**
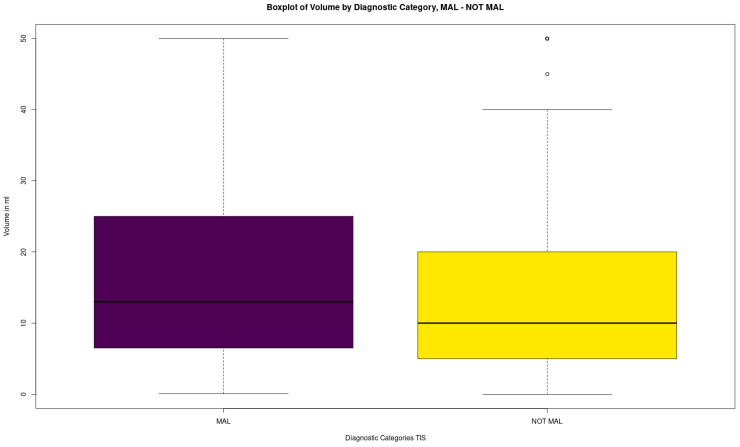
Boxplots of TIS~volume for MAL and NOT MAL.

**Table 1 biomedicines-12-00899-t001:** Number and percentage of serous effusions per TIS category.

	ND	NFM	AUS	SFM	MAL	Total
**Peritoneal**	15 (2.14%)	484 (69.14%)	21 (3%)	20 (2.86%)	160 (22.86%)	700
**Pleural**	66 (4.14%)	1228 (77.04%)	39 (2.45%)	51 (3.2%)	210 (13.17%)	1594
**Pericardial**	5 (10.87%)	27 (58.7%)	0 (0%)	1 (2.17%)	13 (28.26%)	46

**Table 2 biomedicines-12-00899-t002:** Pleural effusions: number of cases, patient gender and age, specimen volume, number of ICC and histology reports for each TIS category.

Diagnostic Category	Gender		Age (Years)	Volume (mL)	ICC	Histology Reports
**ND** (n = 66)	M: 39	F: 27	Min: 11	Min: 1	0	24
			Max: 91	Max: 10		
			Ave: 72.4	Ave: 5.33		
**NFM** (n = 1228)	M: 712	F: 516	Min: 18	Min: 0.5	28	525
			Max: 95	Max: 1600		
			Ave: 69.37	Ave: 182		
**AUS** (n = 39)	M: 10	F: 29	Min: 60	Min: 0.5	7	16
			Max: 83	Max: 50		
			Ave: 75	Ave: 19.38		
**SFM** (n = 51)	M: 33	F: 18	Min: 55	Min: 3	15	40
			Max: 91	Max: 700		
			Ave: 70.2	Ave: 125.57		
**MAL** (n = 210)	M: 99	F: 111	Min: 45	Min: 0.5	126	107
			Max: 95	Max: 1400		
			Ave: 74.3	Ave: 206.5		
**Total** (n = 1594)	M: 893	F: 701	Min: 11	Min: 0.5	176	712
			Max: 95	Max: 1600		
			Ave: 70.92	Ave: 173.75		

**Table 3 biomedicines-12-00899-t003:** Peritoneal effusions: number of cases, patient gender and age, specimen volume, number of ICC and histology reports for each TIS category.

Diagnostic Category	Gender		Age (Years)	Volume (mL)	ICC	Histology Reports
**ND** (n = 15)	M: 10	F: 5	Min: 37	Min: 3	0	4
			Max: 88	Max: 20		
			Ave: 67.75	Ave: 8.7		
**NFM** (n = 484)	M: 262	F: 222	Min: 16	Min: 0.5	28	240
			Max: 89	Max: 2400		
			Ave: 66.47	Ave: 230.44		
**AUS** (n = 21)	M: 11	F: 10	Min: 42	Min: 5	3	11
			Max: 85	Max: 500		
			Ave: 64.6	Ave: 133.2		
**SFM** (n = 20)	M: 11	F: 9	Min: 55	Min: 3	3	17
			Max: 87	Max: 100		
			Ave: 70	Ave: 36.5		
**MAL** (n = 160)	M: 54	F: 106	Min: 35	Min: 1	54	91
			Max: 93	Max: 2000		
			Ave: 70.38	Ave: 245.2		
**Total** (n = 700)	M: 348	F: 352	Min: 16	Min: 0.2	69	363
			Max: 93	Max: 2400		
			Ave: 67.6	Ave: 234.72		

**Table 4 biomedicines-12-00899-t004:** Pericardial effusions: number of cases, patient gender and age, number of ICC and histology reports for each TIS category.

Diagnostic Category	Gender		Age (Years)	ICC	Histology Reports
**ND** (n = 5)	M: 1	F: 4	Min: 54	0	0
			Max: 81		
			Ave: 71.2		
**NFM** (n = 27)	M: 14	F: 13	Min: 25	0	14
			Max: 79		
			Ave: 55.4		
**AUS** (n = 0)	M: -	F: -	Min: -	0	0
			Max: -		
			Ave: -		
**SFM** (n = 1)	M: 0	F: 1	Min: 78	0	0
			Max: 78		
			Ave: 78		
**MAL** (n = 13)	M: 12	F: 1	Min: 50	12	5
			Max: 82		
			Ave: 66.61		
**Total** (n = 46)	M: 27	F: 19	Min: 25	12	19
			Max: 82		
			Ave: 60.78		

**Table 5 biomedicines-12-00899-t005:** Distribution of cases according to TIS in relation to fluid volume.

TIS	ND	NFM	AUS	SFM	MAL	Total
**<10 mL**	4 (1.1%)	298 (81.2%)	9 (2.5%)	14 (3.8%)	42 (11.4%)	367
**10–500 mL**	2 (0.3%)	563 (77.7%)	11 (1.5%)	21 (2.9%)	128 (17.7%)	725
**>500 mL**	0 (0%)	68 (76.4%)	3 (3.4%)	2 (2.3%)	16 (17.9%)	89
**Total No.**	6	929	23	37	186	1181

**Table 6 biomedicines-12-00899-t006:** Distribution of cases according to TIS.

ND	NFM	AUS	SFM	MAL	Total
6 (0.6%)	773 (79.1%)	17 (1.7%)	30 (3.1%)	151 (15.5%)	977

**Table 7 biomedicines-12-00899-t007:** Dunn’s test results (AUS, MAL, ND, NFM, SFM). The adjusted *p*-value is significant for the combination of MAL–NFM.

Comparison	Z	*p*.Unadj	*p*.Adj
1 AUS–MAL	−1.39	0.16	1
2 AUS–ND	1.51	0.13	1
3 MAL–ND	2.46	0.01	0.14
4 AUS–NFM	−0.39	0.7	1
5 MAL–NFM	2.9	0	0.04
6 ND–NFM	−1.92	0.06	0.55
7 AUS–SFM	−0.42	0.68	1
8 MAL–SFM	1.15	0.25	1
9 ND–SFM	−1.85	0.06	0.65
10 NFM–SFM	−0.16	0.87	1

**Table 8 biomedicines-12-00899-t008:** Volumes distribution summaries for MAL and NFM.

	Min.	1st Qu.	Median	Mean	3rd Qu.	Max.
**MAL**	0.10	6.50	13.00	18.94	25.00	50.00
**NFM**	0.01	5.00	10.00	14.60	20.00	50.00

**Table 9 biomedicines-12-00899-t009:** Dunn’s test results (MAL, NFM, Uncertain). The adjusted *p*-value is not significant for the combination of NFM–UNCERTAIN.

Comparison	Z	*p*.Unadj	*p*.Adj
1 MAL–NFM	3.055861	0	0.01
2 MAL–UNCERTAIN	2.750022	0.01	0.02
3 NFM–UNCERTAIN	1.177339	0.24	0.72

**Table 10 biomedicines-12-00899-t010:** Volumes distribution summaries for MAL, NFM, and UNCERTAIN.

	Min.	1st Qu.	Median	Mean	3rd Qu.	Max.
**MAL**	0.10	6.50	13.00	18.94	25.00	50.00
**NFM**	0.01	5.00	10.00	14.60	20.00	50.00
**UNCERTAIN**	0.20	5.00	7.00	13.79	20.00	50.00

**Table 11 biomedicines-12-00899-t011:** Volumes distribution summaries for MAL and NOT MAL.

	Min.	1st Qu.	Median	Mean	3rd Qu.	Max.
**MAL**	0.10	6.50	13.00	18.94	25.00	50.00
**NOT MAL**	0.01	5.00	10.00	14.54	20.00	50.00

## Data Availability

The data presented in this study are available within the article.

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
