# Peer review of "Optimal Volume Assessment for Serous Fluid Cytology"

_biomedicines, 2024, doi:10.3390/biomedicines12040899_

Round 1
Reviewer 1 Report
Comments and Suggestions for Authors
1. It was mentioned in 2.1Patient selection that sample volume (available for 1181 specimens) but in Table 2 only pleural effusion, the total number is 1594 with volume information. (conflicting).
2. It is good to have such large sample size for study. In several past studies, some of them use malignancy prevalence (fraction) in each volume bins to present diagnostic accuracy since pleural biopsy is usually not available for golden standard. In your study per Table 5, combing SFM and MAL rates, volume 10-500 ml and >500 ml have a much higher malignancy rate than <10 ml volume. Maybe need more explanation for no significant difference.
3. Following last paragraph, it is also important to control other variables, such as age, gender, adult or children, any prior cancer history and confirms they are similar between three volume groups.
4. Maybe it is important to explain whether this study can have sufficient power (sample size issue) to detect a significance between specimens in different volume bins.
Author Response
We would like to express our gratitude to the reviewers for their time and efforts in reviewing our manuscript and for their valuable comments. Our specific reply to every comment is as follows:
- Tables 1,2,3,4 and charts 1,2 display our whole dataset of 2340 cases, including the cases with no volume information available. Out of these, 1594 are pleural, 700 are peritoneal and 46 are pericardial serous effusions.
- Indeed when closely looking at table 5, the values presented appear to show higher malignancy rates for specific volumetric bins for specific groups. To assess whether this has statistical significance a statistical test was applied. Since the chi-squared test approximation may be unreliable for contingency tables containing values lower than 5, we used the Fisher’s exact test to get a more accurate result. For this contingency table, a Fisher’s exact test for count data will give the p-value of 0.07828, which is not statistically significant.
- Thank you for this comment. We have added a special mention on that in the paragraph of the limitations (highlighted in the manuscript).
- In this study we use the Kruskal-Wallis test, which is a non-parametric ANOVA test. We have selected to use Cohen's f statistic as the effect size index to use for our Kruskal-Wallis analysis of variance. That way we can measure a standardized average effect in our observations across all the levels of the Volume variable. The calculated value for our dataset was f = 0.1248913. According to Jacob Cohen, the values of 0.10, 0.25, and 0.40 represent small, medium, and large effect sizes, respectively (Cohen, J. (1988). Statistical power analysis for the behavioral sciences. Second Edition. Erlbaum.). The balanced one-way analysis of variance power calculation for significance level 0.05 outputs a power level of 1, denoting a very high confidence level for our sample size of 977 observations (outliers removed). Therefore it appears that our sample size has adequate power to detect a small effect size, which is a strong indication that the study design is well-suited to identify the effect under investigation.

Reviewer 2 Report
Comments and Suggestions for Authors
The study with title "Optimal Volume Assessment for Serous Fluid Cytology", Authored by Konstantinos Christofidis and colleagues investigates the optimal volume of serous fluid required for accurate diagnosis using the International System for Reporting Serous Fluid Cytopathology (TIS), alongside providing insightful information on the distribution of serous effusion cases within TIS categories and relevant epidemiological data. The retrospective analysis of 2340 serous effusion cases from two hospitals offers a comprehensive overview, presenting a clear correlation between fluid volume and TIS categorization.
The text is well-written and easy to follow. However, it would enhance the reader's understanding if the TIS categories were explained briefly in the abstract or omitted, since they are to be detailed in the main text, ensuring clarity and continuity for readers unfamiliar with the terminology.
Additionally, the labeling and explanation of visual aids could be improved. For instance, I would recommend changing "Chart 1" to "Figure 1" and "Chart 2" to "Figure 2". Including more informative legends in the charts anf Figure 1 with brief explanation of the pie charts and boxplots would aid in interpreting the data more effectively.
In conclusion, the study presents a solid analysis with minor areas for improvement in terms of presentation and clarity.
Author Response
We would like to express our gratitude to the reviewers for their time and efforts in reviewing our manuscript and for their valuable comments. Our specific reply to every comment is as follows:
- We agree that the TIS categories should be explained in the abstract. A short explanation for each of them has been added (highlighted).
- The word “Chart” has been changed to “Figure”. Short explanations are included in the legends (highlighted).
